# Green Extraction of Annatto Seed Oily Extract and Its Use as a Pharmaceutical Material for the Production of Lipid Nanoparticles

**DOI:** 10.3390/molecules27165187

**Published:** 2022-08-15

**Authors:** Sônia do Socorro do C. Oliveira, Edmilson dos S. Sarmento, Victor H. Marinho, Rayanne R. Pereira, Luis P. Fonseca, Irlon M. Ferreira

**Affiliations:** 1Laboratório de Biocatálise e Síntese Orgânica Aplicada, Departamento de Ciências Exatas, Universidade Federal do Amapá, Rod. JK, KM 02, Macapa 68902-280, Brazil; 2Instituto de Pesquisas Científicas e Tecnológicas do Estado do Amapá, Macapa 68901-025, Brazil; 3Departamento de Bioengenharia, Institute for Bioengineering and Biosciences (iBB), Instituto Superior Técnico (IST), Universidade de Lisboa, Av. Rovisco Pais 1, 1049-001 Lisbon, Portugal

**Keywords:** Brazil nut oil, *Bixa orellana*, annatto, amazon oils, urucum

## Abstract

This work developd nanomaterials formulated from annatto seed oily extract (ASE), myristic acid (tetradecanoic acid), and their fatty acid esters. The annatto seed oily extract was obtained using only soybean oil (ASE + SO) and Brazil nut oil (ASE + BNO). The UV/VIS analysis of the oily extracts showed three characteristic peaks of the bixin molecule at 430, 456 and 486 nm. The lipid nanoparticles obtained using myristic acid and ASE + BNO or only BNO showed better results than the oil soybean extract, i.e., the particle size was <200 nm, PDI value was in the range of 0.2–0.3, and had no visual physical instability as they kept stable for 28 days at 4 °C. Lipid nanoemulsions were also produced with esters of myristic acid and ASE + BNO. These fatty acid esters significantly influenced the particle size of nanoemulsions. For instance, methyl tetradecanoate led to the smallest particle size nanoemulsions (124 nm), homogeneous size distribution, and high physical stability under 4 and 32 °C for 28 days. This work demonstrates that the chemical composition of vegetable oils and myristic acid esters, the storage temperature, the chain length of fatty acid esters (FAE), and their use as co-lipids improve the physical stability of lipid nanoemulsions and nanoparticles from annatto seed oily extract.

## 1. Introduction

*Bixa orellana* L. is an angiosperm from the Bixaceae Kunth family. It is popularly known as “urucum” (Brazil), “achiote” (Peru and Cuba), “axiote” (Mexico), and “annatto” (USA) [1,2,3]. Annatto offers many health benefits, including anti-cancer and anti-inflammatory properties and protective and preventative action against cardiovascular diseases, cataracts, and macular degeneration [4]. Annatto seed oil is rich in bioactive compounds, such as geranylgeraniol diterpene and tocotrienols, which are used as ingredients in the food industry due to their anti-inflammatory and antioxidant properties [5,6]. The pericarp of these seeds is rich in carotenoids, mainly apocarotenoids such as bixin and norbixin [7,8].

Carotenoids are hydrophobic and have been extracted from raw vegetable materials with organic solvents (alcohol, acetone, chloroform, hexane, ethyl ether, or petroleum ether), which are harmful to human health and the environment [9,10]. Therefore, green and sustainable methods for extracting bioactive and natural products for medicinal, food, and cosmetic use are increasing. Non-toxic vegetable oils are often used in green extraction as they present a lower risk to health, preserve the organoleptic characteristics of the extracted product, do not harm the environment, and are therefore seen as a sustainable, beneficial alternative [10,11]. Palm oil, soybean oil, and sunflower oil, among others, are alternative solvents that have been used in extractive processes [12,13].

Oily extracts possess biological activity intrinsic to their phytochemical constitution and potentially beneficial physicochemical properties for use as functional excipients in pharmaceutical formulations. “Nano” pharmaceutical formulations effectively encapsulate vegetable extracts, improving bioavailability, preventing degradation, and improving biological activity [14]. Nanoemulsions and nanostructured lipid carriers (NLCs) are “nano” pharmaceutical formulations derived from emulsions. They are oil-in-water (O/W) emulsions, and the oil phase consists of a blend of liquid and solid lipids that remains liquid or solid at temperatures above room and body temperature, respectively. While nanoemulsions are O/W or water-in-oil (W/O) emulsions, the oil phase is a liquid lipid. The mean droplet/particle size is usually <500 nm in both nanoemulsions and lipid nanoparticle systems. This small droplet/ particle size gives them a clear or nebulous appearance [15,16].

The lipidic interior of NLCs and nanoemulsions is highly suitable for carrying and transporting lipophilic substances and drugs [17,18]. While NLCs and nanoemulsions produced with vegetable oils have been frequently reported in scientific studies [19,20], there is no record of the use of annatto oil extract as the oil phase of these drug delivery systems.

The present study aims to obtain annatto seed oily extract using Brazil nut oil (*Bertholletia excelsa* Bonpl.) and soybean oil as liquid extractors and potential vehicles of pharmaceutically active compounds in the development of nanoemulsions and NLCs. Additionally, the effect of the composition of the resulting nanoemulsions and NLCs on different storage temperature conditions and the physical properties of formulations containing annatto seed oily extract from soybean or Brazil nut oils and myristic acid and respective esters as co-lipids were evaluated.

## 2. Results

### 2.1. Characterization of Annatto Seed Oily Extract (ASE) 

#### 2.1.1. Fatty Acid Composition of the Vegetable Oils

Soybean oil is commonly used in producing NLCs as it is easily accessible and adequate in the balance of saturated/unsaturated fatty acids favorable to the stabilization process of the NLCs. However, based on Amazon biodiversity, numerous advances have been made in recent years to demonstrate the health benefits of oils from Amazon, and among these, *B. excelsa* seed oil stands out. The oils extracted from the seed of *B. excelsa* show notable high nutritional value and several biological activities, such as healing, antioxidant, and anti-inflammatory activities [21,22,23]. Furthermore, this oil has a high stearic acid content and low concentration of other saturated fats, excellent characteristics for the oil phase in the preparation of NLCs [24]. Stearic acid is also non-atherogenic and does not pose a risk for heart disease when compared to other saturated fatty acids [25]. Therefore, *B. excelsa* seed oil becomes an ideal candidate for preparing NLCs.

The lipid profiles of the oil samples (*B. excelsa* seed oil and soybean oil) are summarized in Table 1. The analysis of the fatty acid composition in the form of ethyl esters of Brazil nut (*B. excelsa*) seed oil and soybean oil was performed by gas chromatography (GCMS-QP 2010) coupled with mass spectrometry in a Shimadzu/GC2010, followed the protocol described by Ferreira et al. [26].

*B. excelsa* seed oil and soybean oil presented similar profiles regarding the identified main fatty acids (FAs). However, the proportions of these FAs varied significantly between the samples. While soybean oil contained mostly saturated fatty acids, such as palmitic acid (11%) and stearic acid (4.55%), the *B. excelsa* seed oil had a saturated fatty acid proportion of 19% and 16% for palmitic and stearic acids, respectively. It is noteworthy that the monounsaturated fatty acid oleic acid (C 18:1, ω-9) was identified as the most significant constituent in *B. excelsa* seed oil samples, accounting for 32% while representing only 22% of soybean oil. In contrast, the polyunsaturated fatty linoleic acid (C 18:2, ω-6) represented a proportion of 54% of soybean oil but only 29% of *B. excelsa* seed oil.

The density of vegetable oils and fats is influenced by the length of the carbon chain and the degree of unsaturation of the fatty acids that compose it. Long FA chain lengths lead to lower density values, while higher degrees of unsaturation increase density value [27]. The density values found herein were 0.8945 mg/mL and 0.8785 mg/mL for *B. excelsa* seed oil and soybean oil, respectively.

#### 2.1.2. Fourier Transform Infrared Spectroscopy Analysis

The ASE-BNO and ASE-SO spectras (Figure 1) showed characteristic bands of triglycerides from vegetable oils. The band at 3013 cm^−1^ was attributed to the C-H stretching vibration of the unsaturated fatty acids. The high-intensity peaks at 2934 cm^−1^ and 2848 cm^−1^ are due to asymmetric and symmetric aliphatic C-H stretchings, respectively [23]. The strong absorption band at 1754 cm^−1^ is assigned to the stretching vibration of the ester carbonyl group in the triglycerides. The bands at 1110 cm^−1^ and 1036 cm^−1^ are ascribed to the stretching vibration of the ester linkage in triacylglycerols [28,29].

### 2.2. Physical Characterization and Evaluation of the Stability of NLCs from Annatto Seed Oily Extract

Vegetable oils have been applied as components of nanoemulsions, lipid nanoparticles, and other systems [19,30,31] because of their diversity of lipid composition and the presence of bioactive compounds. These compounds add medicinal value and functional properties to the NLCs, for instance, *B. excelsa* oil contains a diversity of phenolic compounds, favoring the antioxidant potential of the formulations [9].

NLCs are prepared from O/W emulsions in which the lipid phase is always a blend of solid and liquid lipids. However, the oil phase of NLCs comprises lipids that remain solid at room and body temperature producing nanoparticles [29]. A variety of solid lipids have been used in the production of NLCs, such as vegetable fats, including *ucuúba* fat [18] and *tucumã* fat [20], as well as other fatty acids. Myristic acid is a 14-carbon fatty acid with a melting point of 54 °C and has been reported in the literature as a pharmaceutical ingredient of NLCs, which have been produced with a blend of myristic acid with sunflower oil [32], myristic acid with *Antarctic krill* (*Euphausia Superba*) oil [33], and myristic acid with oleic acid [34].

Myristic acid was used in this work as a solid lipid in four formulations (Table 2). The lipid composition of these formulations was a blend of myristic acid with ASE + BNO, ASE + SO, BNO, and SO (Table 2). Immediately after preparation, the NLCs were visually analyzed, and all were found to be homogeneous, without phase separation or the precipitation of lipids, presenting typical coloration of the lipid phase used. The NLC formulations prepared with ASE had a strong yellow color, typical of annatto extract. The other myristic acid formulations without annatto extract presented a white color (Figure 2).

After the first 24 h, the particle size, PDI, and zeta potential of the myristic acid formulations were also evaluated (Figure 3). On the first day of preparation, all the formulations had particle sizes smaller than 200 nm, except ASE + SO_NLC_ (204.9 ± 2.6 nm). The lipid matrix of the BNO_NLC_ (free of annatto seed extract) had very similar particle sizes to ASE + BNO_NLC_, 171.2 ± 1.77 nm and 173.4 ± 1.45 nm, respectively. In contrast, the formulation with SO_NLC_ (free of soybean oil extract) exhibited a particle size of 191.8 ± 1.59 nm (Figure 3A).

After the first measurement, the NLC formulations were monitored during 28 days of storage at 4 °C and 32 °C. Storage temperature critically influenced the NLC properties. The BNO_NLC_ and ASE + BNO_NLC_ presented the smallest particle size variation at 4 °C when compared with the samples using SO_NLC_ and ASE + SO_NLC_ (Figure 3). From the 7th to the 21st day, the mean particle size values for BNO_NLC_ at 4 °C varied from 137.9 ± 0.68 nm to 162.4 ± 1.5 nm, while ASE + BNO_NLC_ exhibited values ranging from 137.4 ± 0.31 nm to 168.9 ± 1.17 nm. The analyses also demonstrated that, at 4 °C of storage, small variations in size were found for SO_NLC_ (175.3 ± 0.76 to 204.1 ± 5.65) and for ASE + SO_NLC_ (199.7 ± 3.44 nm to 237.2 ± 3.16 nm). However, when the samples were stored at 32 °C, larger variations in particle size were found for all the NLC formulations (Figure 3). Furthermore, from the 7th day onwards, a slight creaming was detected in the samples prepared with soybean oil, stored at 32 °C.

Concerning PDI on the first day, i.e., after the preparation of the samples, the values obtained for all the NLCs varied from 0.2 to 0.3 (Figure 3). The PDI value estimates the average uniformity of nanoparticle size, as larger PDI values represent broad particle size distribution. A PDI value lower than 0.3 indicates a relatively narrow particle size distribution, and the lower the value, the more monodisperse the system [29,35,36].

The PDI values for BNO_NLC_ remained stable at 0.2 from the 7th to the 28th day, under both storage temperatures, indicating that the size distributions were suitable and uniform in the NLC systems. The ASE + BNO_NLC_ samples exhibited values ranging from 0.3 to 0.4 PDI (4 °C) and 0.3 to 0.5 PDI (32 °C), with higher PDI values found on the 28th day. Meanwhile, the formulations prepared using SO_NLC_ and ASE + SO_NLC_ exhibited PdI values ranging from 0.3 to 0.4 (4 °C) and 0.2 to 0.3 when the temperature increased (32 °C) (Figure 3B). The PDI values show that only one formulation was polydispersed (ASE + BNO_NLC_ at 32 °C, PDI 0.5), with PDI values >4 considered highly polydisperse [37]. Finally, on the 28th day of storage at under 4 °C, the NLCs showed the following results: ASE + BNO_NLC_ 167.4 ± 3.41 nm, BNO_NLC_ with 134.1 ± 0.7 nm, SO_NLC_ 168.8 ± 4.42, and ASE + SO_NLC_ 223.5 ± 2.35 nm (Figure 3A).

The significant changes in PDI and particle size values were observed only in formulations stored at a temperature of 32 °C, which leads to the inference that temperature influences the physical stability of the formulations. This instability of NLCs at 32 °C is probably due to the higher kinetic energy of the system leading to increased collisions between particles, the consequence of which is aggregation and an increase in particle size [38]. However, even though the changes occur only at 32 °C, the possibility that the changes in particle size occur because of Ostwald ripening cannot be ruled out. The increase in particle size of monodisperse formulations is a characteristic of systems that suffer from Ostwald ripening [39]. The process by which larger particles grow while smaller particles reduce is due to the mass transport of the dispersion phase through the activated continuous intervening phase by Laplace pressure [19,39]. 

Finally, with respect to the zeta potential measurements on the 1st day, the mean zeta potential values for all formulations ranged from −58.2 ± 1.38 to −65.6 ± 0.26 mV (Figure 3). The NLCs (free of ASE) had the highest zeta potential values on the 1st day after preparation (−65.6 ± 0.26 mV for BNO_NLC_ and −58.2 ± 1.38 mV for SONLC) (Figure 3B). The NLCs with ASE had values of −37.3 ± 1.15 mV and −37.7 ± 1.5 mV for ASE + SO_NLC_ and ASE + BNO_NLC_, respectively (Figure 3).

The zeta potential value is an important parameter in the characterization of nanoparticle systems. In general, nanomaterials with a zeta potential above or below ±30 mV are physically stable, as higher surface charges lead to a stronger repulsion effect between dispersing nanoparticles and, therefore, greater stability. Furthermore, negative zeta potential values are due to the presence of terminal carboxylic groups in lipids [40,41].

Additionally, with respect to the zeta potential measurements on the 7th day after preparation, the BNO_NLC_ at 32 °C and 4 °C showed values of −55 ± 2.44 mV and −44.3 ± 2.34 mV, respectively (Figure 3B). All samples exhibited promising zeta potential values from the 14th to the 21st day of storage, with no significant variations. These results show that the lipid matrix and/or the presence of the oil extract did not significantly influence the zeta potential values over time at the different storage temperatures studied (Figure 3B).

### 2.3. Physical Properties and Evaluation of the Stability of Nanoemulsions (Nn) Based on Myristic Acid Esters (Co-Lipids)

The results of the myristic acid formulations show that the NLCs with BNO had smaller particle sizes and small size distribution according to PDI values and that there was no visual sign of physical instability due to high zeta potential values. Thus, the ASE + BNO were used to compose the formulations produced from myristic acid esters. 

New colloidal systems based on myristic acid esters (methyl tetradecanoate (MT), ethyl tetradecanoate (ET), isopropyl tetradecanoate (IT), and butyl tetradecanoate (BT)) were produced, and evaluated. The esters of long-chain fatty acids contribute to the formation of nanoemulsions (*Nn*), i.e., nano-liquid or semi-solid droplets, and improve their encapsulation capacity and stability of *Nn* by steric interaction, hydration in the surface layer, and resistance to flocculation and coalescence preparation of *Nn* [42]. For example, a mixture of Precirol ATO 5 and a fatty alcohol or fatty acid ester was a successful strategy for *Nn* production [43].

The myristic acid esters used in the production of these new *Nn* formulations were MT (methyl tetradecanoate), ET (ethyl tetradecanoate), IT (isopropyl tetradecanoate), and BT (butyl tetradecanoate). The length of the carbon chain of these esters grows in the following order: MT < ET < IT < BT, and at room temperature, all these colloidal materials are liquids in opposition to respective myristic acid that is solid. The results obtained from myristic acid esters led to the production of nanoemulsions. Unlike NLC, nanoemulsions are emulsified systems in which the oil phase consists of liquid lipids, i.e., not generating lipid solid particles [19].

The *Nn* formulations were also characterized in terms of particle size, zeta potential, and PDI. Figure 4 shows the differences in nanoemulsion size according to the carbon chains of the esters and their influence on the other physicochemical parameters. The size of the carbon chains of the myristic acid esters significantly influenced (*p* < 0.05) the droplet size of the nanoemulsion (Figure 4). The particle size and the carbon chain size of the ester were related, with methyl tetradecanoate having the smallest particle size (124.5 ± 0.95 nm), followed by ethyl tetradecanoate (144.2 ± 2.00 nm), isopropyl tetradecanoate (153.4 ± 0.74 nm), and butyl tetradecanoate (192.8 ± 4.5 nm). The increase in the carbon chain is related to the increased hydrophobicity of the lipid, which results in a larger droplet size [32].

However, no correlation or no significant difference was observed between the number of the carbon chain of the esters and the PDI values (Figure 4). The PDI values found were in the range of 0.2–0.4, characterizing the nanoemulsions obtained with the methyl tetradecanoate (0.4 ± 0.01), ethyl tetradecanoate (0.329 ± 0.01), isopropyl tetradecanoate (0.278 ± 0.03), and butyl tetradecanoate (0.304 ± 0.01 nm) as monodisperse.

All the formulations presented zeta potential values below −30 mv (Figure 4), while, as with the particle size, these values also varied according to the carbon chain of the esters (Figure 4).

As the nanoemulsions coupled with methyl tetradecanoate were the smallest, they were chosen for the study of physical stability. The nanoemulsions were evaluated at two different temperatures (4 and 32 °C) for 28 days. Interestingly, from the 7th day onwards, a considerable reduction in the particle size and PDI of (ASE + BNO)_Nn-MT_ was observed at both storage temperatures, which remained practically constant until the 28th day (Figure 5A,B). The control of nanoparticle size is essential during storage, as it allows the physical stability of the formulation to be verified and confirmed [44,45].

The decrease in the size values observed between the first and the seventh day is possibly a consequence of the method of nanoemulsion production, where there was a slight interaction between nano-droplets that was gradually reduced with storage time. This observation in this study is in total opposition with the typical process of physical instability of emulsified systems, where the size of formulation increases due to flocculation, coalescence, or Ostwald ripening. Usually, flocculation and coalescence lead to a wide droplet size distribution with a high PDI [39]. PDI and droplet size did not increase during storage, not even when subjected to high temperatures (32 °C). This observation leads us to conclude that these nanoemulsion formulations based on myristic acid esters are stable and do not undergo flocculation or coalescence (Figure 5). Ostwald ripening appears with an increase in droplet size, even when the PDI remains at low values characteristic of monodisperse systems [19,39]. Figure 5 shows that PDI and droplet size values do not characterize Ostwald ripening since no nanoemulsion droplet growth exists.

The zeta potential of (ASE + BNO)_Nn-MT_ at both storage temperatures (4 °C and 32 °C) decreases slightly from 0 to 7 days after preparation. This decrease in the zeta potential increases the repulsion between the droplets and makes their aggregation difficult, as proved, in this work, by reducing droplet size during the same period of storage time (Figure 5). The zeta potential of the formulations prepared by Makoni et al. [38] also showed a reduction in the first week of storage but remained constant for 8 weeks. The decrease in the zeta potential increases the repulsion between the droplets and minimizes aggregation, as proved by the droplet size reduction in this work. 

### 2.4. UV-VIS Characterization of Annatto Nn and NLC Formulations

Figure 6 illustrates the spectral profile of nanoemulsion formulations containing bixin extract diluted in chloroform. In all the formulations, the expression of three peaks characteristic of the bixin molecule occurred at 430, 456 and 486 nm. In the study of the interaction of bixin with aprotic solvents, [46] showed that the maximum absorption of bixin occurs in the second characteristic peak of this substance (456 nm), corroborating the UV-VIS spectra obtained. However, for the NLC from soybean oil, there was distortion and displacement of the bands to a higher energy region. In this case, the change in UV-VIS absorption may be related to the aggregation of carotenoids to the bioactive compounds present in the composition of the lipid phase and/or photodegradation action on carotenoids. Unlike a higher wavelength shift, a blue or hypsochromic shift is observed, where the absorption band shifts to a lower wavelength. The type of aggregation observed in dyes that results in a blue shift is referred to as H-aggregate (a shift to a shorter wavelength, hypsochromic).

## 3. Materials and Methods

### 3.1. Materials

Annatto seeds of the yellow phenotype were obtained from the state of Amapá in the Amazon region of Brazil (N 00° 15,918′ W 051° 08,141′) in September 2020. The myristic acid (99%) was purchased from Sigma-Aldrich (São Paulo, Brazil), the Brazil nut oil (BNO) from Comaja^®^ (Laranjal do Jari, Brazil), and the Brazil and the soybean oil (SO) from Concordia^®^ (São Paulo, Brazil). The surfactant Tween 80^®^ (poly-oxyethylene sorbitan monooleate, HBL 15.0) was obtained from Synth-Aldrich (São Paulo, Brazil). For the aqueous phase of the formulations, water for injection was purchased from Fresenius Kabi Ltd.a (São Paulo, Brazil). Oxone^®^ (KHSO_5_.0.5KHSO_4_.0.5K_2_SO_4_) was purchased from Sigma-Aldrich (São Paulo, Brazil). Methanol (99.8%), ethanol (98%), isopropanol (98%), isobutanol (98%), *n*-hexane (98.5%), and ethyl acetate (99.5%) were all purchased from Synth (São Paulo, Brazil).

### 3.2. Production of Annatto Seed Oily Extract (ASE)

To obtain the annatto seed oily extract (ASE), we followed the protocol described by Rios and Mercadante [47], with some modifications. Whole annatto seeds *in natura* (15 g) underwent maceration with a 100 mL lipid medium, 100 mL of Brazil nut seed oil (BNO) or soybean oil (SO), under magnetic stirring (400 rpm), overnight, until the complete removal of the pigment, followed by filtration. The oily extracts collected, soybean oily extract from annatto seeds (ASE + SO) and Brazil nut oily extract from annatto seeds (ASE + BNO) were stored in amber glasses, protected from light and kept refrigerated (4 °C) for later use.

### 3.3. Synthesis of Methyl, Ethyl, Isopropyl, and Butyl Esters from Myristic Acid Catalyzed by Oxone Salt

For the esterification of the myristic acid, Oxone^®^ was used as a catalyst, while methyl, ethyl, isopropyl, and *n*-butyl alcohols were separately used to produce the respective esters: MT (methyl tetradecanoate), ET (ethyl tetradecanoate), IT (isopropyl tetradecanoate), and BT (butyl tetradecanoate) [48]. Esterification was performed in a 50 mL reaction flask containing 1.0 g of myristic acid, 3 mL of alcohol (separately), and 0.2 g (20%) of Oxone^®^ salt and magnetically stirred for 12 h at 40 °C. Afterwards, the reactions were filtered, the organic phases were dried with anhydrous sodium sulfate and filtered, and the solvents were removed by vacuum under reduced pressure. The ester products were purified by silica gel column chromatography, with a mobile phase comprising a mixture of *n*-hexane and ethyl acetate (9:1). GC-MS characterized the isolated products, according to Ferreira et al. [26].

### 3.4. Fourier Transform Infrared Spectroscopy Analysis

A Fourier transform infrared spectrophotometer (Spectrum Two FT PerkinElmer, Inc., Waltham, MA, USA) with an attenuated total reflection (ATR) sampling accessory, a diamond plate and a deuterated triglycine sulfate detector were used to record the spectra of the ASE + BNO, ASE + SO, BNO, and SO. The spectral range was set between 350 and 4.000 cm^−1^, and the resolution was set to 0.5 cm^−1^ [49].

### 3.5. Preparation of Nanostructured Lipid Carriers (NLC) from Annatto Seed Oily Extract 

The NLCs were prepared according to Pinto et al. [50], with some adaptations. Initially, four formulations were prepared: ASE + BNO_NLC_, ASE + SO_NLC_, BNO_NLC_, and SO_NLC_ (Table 2). For all formulations, the aqueous phase consisted of surfactant Tween 80 (2.5 wt%) dissolved in water for injection (93.5 wt%). The lipid phase consisted of a blend of myristic acid (1.6 wt%), the solid lipid, and oily extracts from annatto seeds with Brazil nut seed oil (ASE + BNO) and soybean oil (ASE + SO); the Brazil nut seed oil (BNO); and the soybean oil (SO) (2.4 wt%), as the liquid lipids. All the NLCs were prepared at a final mass of 10 g (Table 2).

The lipid phase was heated at 64 °C, 10 °C above the melting point of the myristic acid (T_m_~54 °C) to prevent the lipid memory effect, using magnetic stirring at 300 rpm until a clear, uniform oil phase was obtained. The aqueous phase, at the same temperature and with magnetic stirring, was added to the oil phase 15 min before the end of the process at 45 min. After this, the mixture was fully homogenized by magnetic stirring at 3800 rpm for 10 min. The resultant NLCs were further cooled by stirring at room temperature, protected from light, and stored under refrigeration. 

### 3.6. Preparation of Nanoemulsions with Myristic Acid Esters (Co-Lipids) 

The nanoemulsions (*Nn*) were prepared using different myristic acid esters (methyl tetradecanoate (MT), ethyl tetradecanoate (ET), isopropyl tetradecanoate (IT), and butyl tetradecanoate (BT)). A total of five formulations were produced, all of which underwent an aqueous phase as described in Section 3.5. The lipid phases of the *Nn* formulations are denominated (ASE + BNO)_Nn-MT_, (ASE + BNO)_Nn-ET_, (ASE + BNO)_Nn-IT_, and (ASE + BNO)_Nn-BT_ resulting in a blend of (ASE + BNO) (2.4 wt%) and the myristic acid esters (1.6 wt%) described in item 2.3. The nanoemulsion was prepared to a final mass of 10 g (Table 3).

### 3.7. NLC and Nn Characterization 

#### 3.7.1. Physical and Visual Evaluations

The physical characteristics of the NLC and *Nn* formulations were evaluated after 24 h and 7, 14, 21 and 28 days after preparation. Additionally, all the formulations were stored at different temperatures, 4 °C and 32 °C, and the effects of storage on the characteristics and properties of NLC and *Nn* formulation were monitored. Particle size, PDI, zeta potential, and visual appearance (signs of creaming, flocculation, and phase inversion) were analyzed.

#### 3.7.2. Particle Size Distribution 

The NLC and *Nn* formulations were characterized by DLS (dynamic light scattering) using a Malvern Zetasizer Nano (Malvern Instruments, Worcestershire, UK). DLS analysis was performed by applying non-invasive Back-Scatter (scattering angle 173°) at a temperature of 25 °C. Before measuring, the NLC and Nn formulations were diluted (1:10) with water for injection. The results of the average particle size (Z-average) and polydispersity index (PDI) are expressed as the average ± standard deviation of three different sets of measurements for each formulation.

#### 3.7.3. Zeta Potential

Zeta potential was measured by electrophoretic mobility using a Malvern Zetasizer Nano (Malvern Instruments, Worcestershire, UK). The samples were diluted in water for injection (1:10, *v*/*v*) and the measurements were performed in triplicate at 25 °C. 

### 3.8. UV-VIS Characterization of Annamato Seed Oil Formulations

A double-beam Shimadzu UV-Mini-visible spectrophotometer was used to record the absorption spectra over 300–600 nm wavelengths at normal pressure and room temperature. Quartz cuvettes with a 1 cm path length were used for measurements in the solution. The estimated experimental error was 1 nm on the maximum band. A quantity of 5 mg/L solution of the sample (ASE + BNO and ASE + SO) was prepared in chloroform. All the experiments were performed in a room with the lights turned off and the blinds lowered.

### 3.9. Statistical Procedures

All tests were performed in triplicate, and the results were expressed as mean ± standard deviation. The statistical analysis was performed using one-way ANOVA, and *p* < 0.05 was considered statistically significant. All analyses were performed using the GraphPad Prism 8.0 software package (GraphPad Prism Inc., San Diego, CA, USA).

## 4. Conclusions

Annatto seed oily extract (ASE) was obtained using soybean oil and Brazil nut oil as solvents. The characterization of the extracts by FTIR showed that, in both oils, there is an expression of typical peaks of functional groups present in triglycerides, which are the substances in greater proportion in the extracts, which explains why the FTIR bands are mostly triglyceride functional groups.

The oily extract of annatto seeds presented itself as a promising excipient in the development of colloidal systems. The ASE + BNO_NLC_ and BNO_NLC_ formulations displayed better results than the NLCs with ASE + SO or SO. Therefore, ASE + BNO was used to obtain nanoemulsions with myristic acid esters. Methyl tetradecanoate nanoemulsions had smaller particle sizes, PDI, and better physical stability.

When analyzed by UV-VIS, all formulations with ASE in their composition exhibited absorption bands characteristic of bixin, the major carotenoid of annatto. The nanoemulsions obtained with methyl tetradecanoate combined with ASE + BNO showed more adequate stability and interesting physical characteristics with great potential for its application in cosmetic and pharmaceutical products.

## Figures and Tables

**Figure 1 molecules-27-05187-f001:**
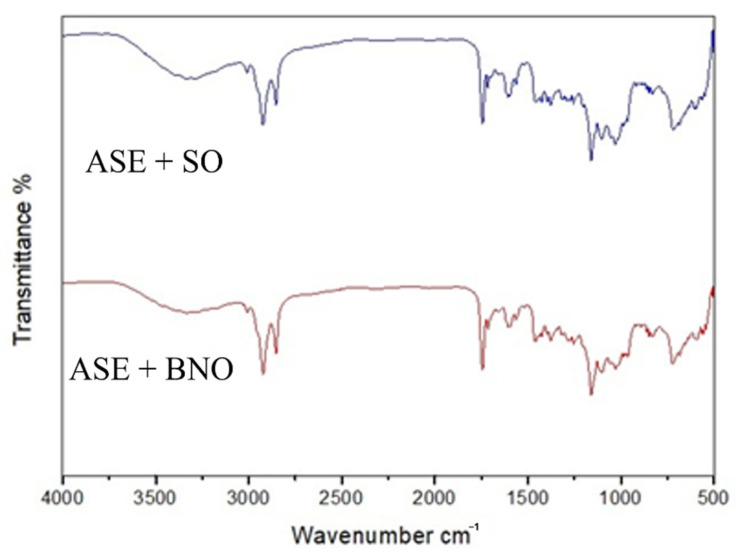
FT-IR spectrum ASE + SO and ASE + BNO.

**Figure 2 molecules-27-05187-f002:**
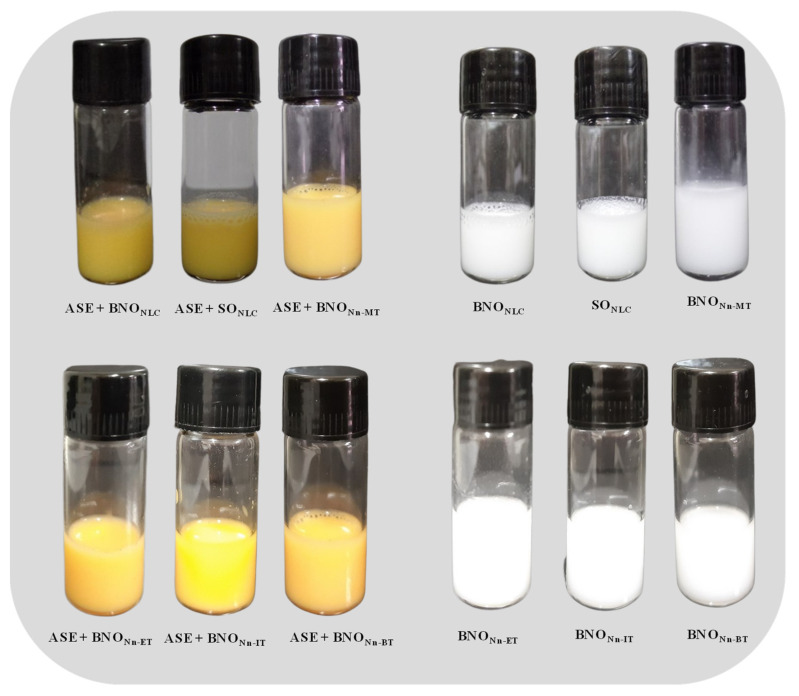
Visual observation of the formulations.

**Figure 3 molecules-27-05187-f003:**
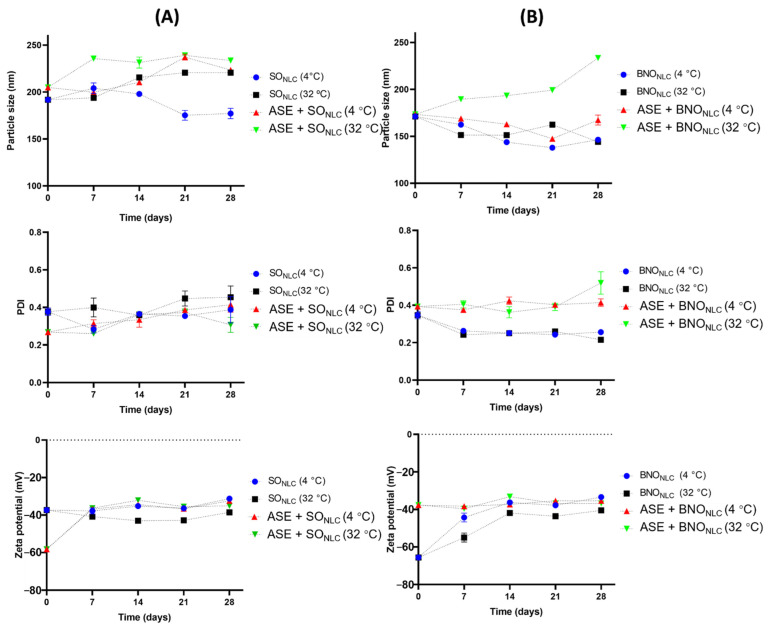
(**A**) Particle size, PDI, and zeta potential of the SO_NLC_ and ASE + SO_NLC_ stored at temperatures of 4 °C and 32 °C. (**B**) Particle size, PDI, and zeta potential of the BNO_NLC_ and ASE + BNO_NLC_ stored at temperatures of 4 °C and 32 °C.

**Figure 4 molecules-27-05187-f004:**
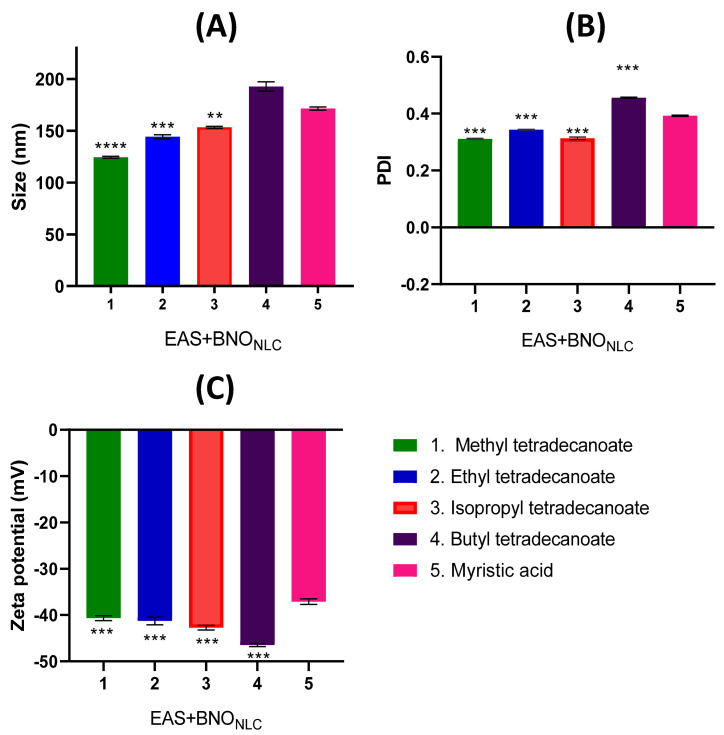
Effect of myristic acid esters on Size (**A**), PDI (**B**), and zeta potential (**C**).

**Figure 5 molecules-27-05187-f005:**
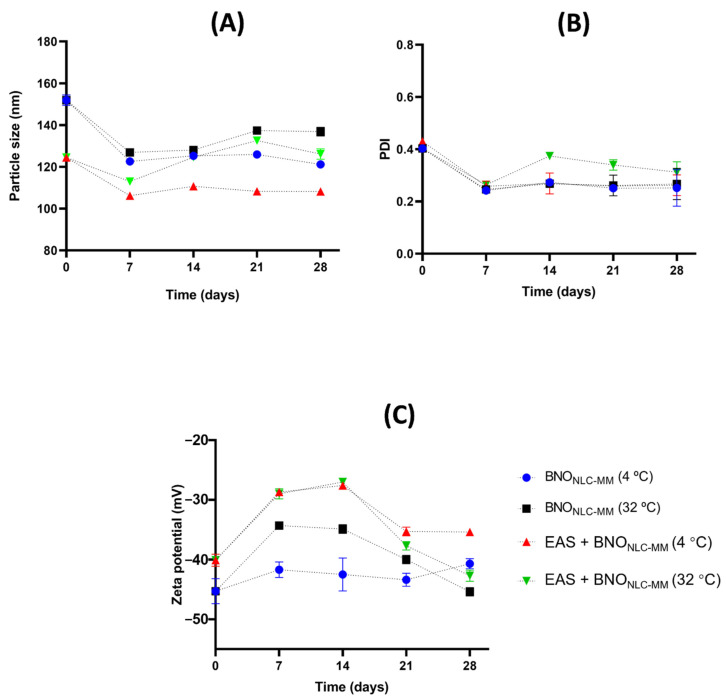
Particle size of SO_NLC_ and ASE + SO_NLC_ (**A**); PDI (**B**) and Zeta potential (**C**) of BNO_NLC_ and ASE + BNO_NLC_ stored at temperatures of 4 °C and 32 °C.

**Figure 6 molecules-27-05187-f006:**
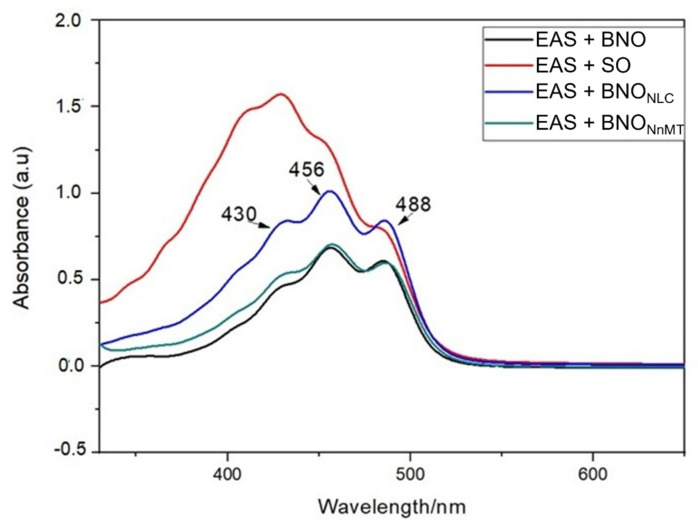
UV-VIS spectra of nanoemulsions formulations and NLCs containing annatto seed oily extract in chloroform at room temperature.

**Table 1 molecules-27-05187-t001:** Mean of fatty acid composition (%) and density of the vegetable oils (Brazil nut and soybean oils) used in the NLC formulations.

Oil			Fatty Acid (%)			Density (g/mL) *	
	Palmitic (C16:0)	Linoleic (C 18:2 ω-6)	Oleic (C 18:1 ω-9)	Linolenic (C 18:1 ω-7)	Stearic (C 18:0)	Unidentified	
Brazil nut	19	29	32	2	16	2	0.89456 (±0.008)
Soybean	11	54	22	1.2	4.55	7.25	0.87850 (±0.048)

* Density: mean ± standard deviation.

**Table 2 molecules-27-05187-t002:** Composition of NLCs from annatto seed oily extract.

Sample	Total Lipids, wt%	Tween 80, wt%	Water, wt%
ASE + BNO	4.0	2.5	93.5
ASE + SO	4.0	2.5	93.5
BNO_NLC_	4.0	2.5	93.5
SO_NLC_	4.0	2.5	93.5

**Table 3 molecules-27-05187-t003:** Composition of nanoemulsions (*Nn*) with myristic acid esters (co-lipids).

Sample	Lipid, wt%	Ester, wt%	Tween 80, wt%	Water, wt%
(ASE + BNO)_Nn-MM_	2.4	1.6	2.5	93.5
(ASE + BNO)_Nn-EM_	2.4	1.6	2.5	93.5
(ASE + BNO)_Nn-PM_	2.4	1.6	2.5	93.5
(ASE + BNO)_Nn-BM_	2.4	1.6	2.5	93.5
(BNO)_Nn-MM_	2.4	1.6	2.5	93.5

## Data Availability

Not applicable.

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
