# Peer review of "Green Extraction of Annatto Seed Oily Extract and Its Use as a Pharmaceutical Material for the Production of Lipid Nanoparticles"

_molecules, 2022, doi:10.3390/molecules27165187_

Round 1

Reviewer 1 Report

In this manuscript, authors extracted annatto seed oily using soybean oil and Brazil nut oil and formulated lipid nanoparticles. The manuscript was written well, and the study was conducted with limited analysis. Following are my queries and suggestion:
·       Author can brief the maceration technique in the introduction and why this conventional technique was chosen for this study.
·       Author can emphasize the novelty of this work for the benefit of readers.
·       Why two set of formulations were developed: one with myristic acid and other with esters myristic acid.  

·       Section 2.3. heading was given as “Physical characterization and evaluation of stability of NLC”, and in materials and method section it was mentioned as “Preparation of nanoemulsions with fatty acid esters (co-lipids)”. Authors can maintain uniformity with NLC.

·       Fig 5 A: What is the potential mechanism behind the decrease in particle size and reduction in zetapotential from 7th day. Looks the sample undergone flocculation during storage. 

Author Response

Molecules

We are very pleased to know that our submitted manuscript entitled “Green extraction of annatto seeds oily extract and use as a pharmaceutical material for the production of lipid nanoparticles. We appreciate the time and effort that you and the reviewers dedicated to providing feedback on our manuscript and are grateful for the insightful comments on and valuable improvements to our paper. We have incorporated all of the suggestions made by the reviewers. Those changes are highlighted within the manuscript. Please see below, in yellow, for a point-by-point response to the reviewers’ comments and concerns

  1. Author can brief the maceration technique in the introduction and why this conventional technique was chosen for this study.

Response: The maceration technique was carried out with 100 mL of lipid medium, 100 mL of Brazil nut seed oil (BNO) or soybean oil (SO), under magnetic stirring (400 rpm), overnight, until the complete removal of the pigment. This conventional technique was chosen for this study because it is low cost, uses accessible equipment, and is close to the technique used by traditional populations, who obtain the extracts through the maceration of seeds in oil, by mechanical friction, in PET bottles, and mainly because bixin is bound to a lipid layer and mechanical friction facilitates its removal.

  1. Author can emphasize the novelty of this work for the benefit of readers.

Response: Soybean oil is commonly used in producing NLCs as it is easily accessible and adequate in the balance of saturated/unsaturated fatty acids favorable to the stabilization process of the NLCs. However, based on Amazon biodiversity, numerous advances have been made in recent years to demonstrate the health benefits of oils from Amazon, and among these, B. excelsa seed oil stands out. The oils extracted from the seed of B. excelsa show notable high nutritional value and several biological activities such as healing, antioxidant, and anti-inflammatory activities [21–23]. Furthermore, this oil has high stearic acid content and low concentration of other saturated fats, excellent characteristics for the oil phase in the preparation of NLCs [24]. Stearic acid is also non-atherogenic and does not pose a risk for heart disease when compared to other saturated fatty acids[25]. Therefore, B. excelsa seed oil becomes an ideal candidate for preparing NLCs (Lines 75 – 85).

  1. Why two set of formulations were developed: one with myristic acid and other with esters myristic acid.  

Response: There was a mistake in the nomenclature used in that item, the correct one is Nn and not NLC, so it is not possible to standardize, since the esters are liquid or semi-solid at room temperature, so we have a liquid lipid phase and no more a mixture of solid and liquid lipids. Nanoemulsions have a liquid oil phase, while NLCs have a solid matrix obtained through a blend of liquid and solid lipids, which will remain solid at room and body temperature.

It is also possible to find such a difference between both formulations of line 50 to 57.

The use of acid myristic esters in this study was informed in text (line 208-214).

Response: New colloidal systems based on myristic acid esters [(methyl tetradecanoate (MT), ethyl tetradecanoate (ET), isopropyl tetradecanoate (IT) and butyl tetradecanoate (BT)] were produced, and evaluated. The esters of long chain fatty acids contribute to the formation of nanoemulsions (Nn), i.e., nano-liquid or semi-solid droplets, and improve their encapsulation capacity and stability of Nn by steric interaction, hydration in the surface layer, and resistance to flocculation and coalescence preparation of Nn[42]. For example, a mixture of Precirol ATO 5 and a fatty alcohol or fatty acid ester was a successful strategy for Nn production [43].

Fig 5 A: What is the potential mechanism behind the decrease in particle size and reduction in zetapotential from 7th day. Looks the sample undergone flocculation during storage. 

Response = The reduce zeta potential and particle size was explained in text (line 250- 272).

The decrease in the size values observed between the first and the seventh day is possibly a consequence of the method of nanoemulsion production, where there was a slight interaction between nano-droplets that was gradually reduced with storage time. This observation in this study is in total opposition with a typical process of physical instability of emulsified systems, where the size of formulation increases due to flocculation, coalescence, or Ostwald ripening. Usually, flocculation and coalescence lead to a wide droplet size distribution with a high PDI [39]. PDI and droplet size did not increase during storage, not even when subjected to high temperatures (32ºC). This observation leads us to conclude that these nanoemulsion formulations based on myristic acid esters are stable and do not undergo flocculation or coalescence (Figure 5). Ostwald ripening appears with an increase in droplet size, even when the PDI remains at low values characteristic of monodisperse systems [19,39]. Figure 5 shows that PDI and droplet size values do not characterize Ostwald ripening since no nanoemulsion droplet growth exists.

The Zeta Potential of (ASE + BNO)Nn-MT at both storage temperatures (4º and 32ºC) decreases slightly from 0 to 7 days after preparation. This decrease in the zeta potential increases the repulsion between the droplets and makes their aggregation difficult, as proved, in this work, by reducing droplet size during the same period of storage time (Figure 5). The zeta potential of the formulations prepared by Makoni et al., [38] also showed a reduction in the first week of storage but remained constant for 8 weeks. The decrease in the zeta potential increases the repulsion between the droplets and minimizes aggregation, as proved by the droplet size reduction in this work.

Reviewer 2 Report

The authors studied seed oily extract using Brazil nut oil (Bertholletia excelsa Bonpl.) and soybean oil as an extractor liquid, applying these extracts as an input pharmaceutical material for nanoemulsions and NLC development, as well as evaluating the effects of different storage temperature conditions on the physical properties of formulations with annatto seed oily extract from soybean or Brazil nut oils and fatty acid esters as co-lipids.

It is a well-written manuscript with which I have no major comments. I only recommend the authors to describe more in the conclusion and emphasize the possibilities of using the NEs developed by them. It is important to emphasize to the readers the practical use of the results.

Author Response

(The authors gave the same response as above.)

Round 2

Reviewer 1 Report

Authors modified the manuscript incorporating the response to queries.